Identification and expression analysis of xyloglucan endotransglucosylase/hydrolase (XTH) family in grapevine (Vitis vinifera L.)

Qiao Tian 1
Zhang Lei 1
Yu Yanyan 1
Pang Yunning 1
Tang Xinjie 1
Wang Xiao 1
Li Lijian 1
Li Bo 2 sdtalibo@163.com
Sun Qinghua 1 qhsun@sdau.edu.cn
1 State Key Laboratory of Crop Biology, College of Life Science, Shandong Agricultural University , Taian, Shandong , China
2 Shandong Academy of Grape, Shandong Academy of Agricultural Sciences , Jinan, Shandong , China
Singh Kashmir
Electronic publication date: 2022 Jun 13
Publication date: 2022
Volume: 10
Electronic Location ID: e13546
Received 2022 Jan 4; Accepted 2022 May 16
Copyright: © 2022 Qiao et al.
Copyright year: 2022
Copyright holder: Qiao et al.
License: This is an open access article distributed under the terms of the Creative Commons Attribution License, which permits unrestricted use, distribution, reproduction and adaptation in any medium and for any purpose provided that it is properly attributed. For attribution, the original author(s), title, publication source (PeerJ) and either DOI or URL of the article must be cited.
License URL: https://creativecommons.org/licenses/by/4.0/

Keywords: VvXTH, Vitis vinifera, Phylogenetic analysis, Expression pattern, Abiotic stress

Funding: National Natural Science Foundation of China 31972358 Natural Science Foundation of Shandong Province, China ZR2018MC022 Shandong Provincial Key Research and Development Project 2019JZZY010727 and 2019GNC106147 The present study was supported by the National Natural Science Foundation of China (31972358), the Natural Science Foundation of Shandong Province, China (ZR2018MC022), and the Shandong Provincial Key Research and Development Project (2019JZZY010727 and 2019GNC106147). The funders had no role in study design, data collection and analysis, decision to publish, or preparation of the manuscript.

==============================
Xyloglucan endotransglucosylases/hydrolases (XTH) are key enzymes in cell wall reformulation. They have the dual functions of catalyzing xyloglucan endotransglucosylase (XET) and xyloglucan endonuclease (XEH) activity and play a crucial role in the responses against abiotic stresses, such as drought, salinity, and freezing. However, a comprehensive analysis of the XTH family and its functions in grapevine (Vitis vinifera L.) has not yet been completed. In this study, 34 XTHs were identified in the whole grapevine genome and then named according to their distribution on chromosomes. Based on a phylogenetic analysis including Arabidopsis XTHs, the VvXTHs were classified into three groups. Cis-element analysis indicated that these family members are related to most abiotic stresses. We further selected 14 VvXTHs from different groups and then examined their transcription levels under drought and salt stress. The results indicated that the transcription levels of selected VvXTHs in the leaves and roots presented the largest changes, suggesting that VvXTHs are likely to take part in the responses to drought and salt stress in grapevines. These results provide useful evidence for the further investigation of VvXTHs function in response to abiotic stresses in grapevine.

Introduction

As one of the most economically fruit crops, grapevine (Vitis vinifera L.) is cultivated worldwide (Feng, He & Hirotomo, 2000). However, the growth of grapes in natural environment is inevitably impacted by a series of abiotic stresses, including salinity, drought, and extreme temperatures, which damage the cell walls of the plants, disrupt the biofilm system, and ultimately affect the quality and yield of the fruit (Araujo, Abiodun & Crespo, 2016; Liu et al., 2019; Ning et al., 2017). Xyloglucan endotransglucosylase/hydrolase (XTH) can carry out cell wall structural modification and rearrangement by severing and repolymerizing cellulose mono-xyloglucan cross-linked structures (Campbell & Braam, 2010). It belongs to the glycoside hydrolase 16 (GH16) family, which is a subfamily of glycoside hydrolases containing diverse enzymes with different specific targets, such as keratan sulfate, β-1,3-glucans, mixed linkage β-1,3(4)-glucans, xyloglucans, j-carrageenan, and agarose (Mark et al., 2009; Stratilova et al., 2020). All XTH proteins present the typical structure of XTH enzymes: the (D/N)E(I/L/A/V/F)(D/T)(F/I)E(F/I/L)LG motif, which includes the catalytic active-site residues ExDxE (Matsui et al., 2005; Liu et al., 2007; Miedes & Lorences, 2009; Singh et al., 2011). XTH proteins may present one or two enzyme activities: xyloglucan endonuclease (XEH) activity and/or xyloglucan endotransglucosylase (XET) activity. The former specifically hydrolyzes xyloglucan β-1,4 glycosidic bonds, cleaving the xyloglucan chain, thereby shortening the xyloglucan chain, and the latter can transfer xyloglucan fragments between xyloglucan chains, elongating xyloglucan chains (Han et al., 2016).

Thus far, the XTH family has been identified and analyzed in species such as Arabidopsis thaliana (33), Hordeum vulgare (24), Glycine max (61), and Nicotiana tabacum (54). (Nomchit et al., 2010; Li et al., 2018; Meng et al., 2018; Cheng et al., 2021; Strohmeier et al., 2004; Tiika et al., 2021). The XTH family was initially classified into three groups, named groups I, II, and III, in Arabidopsis (Campbell & Braam, 1999). However, a subsequent study in Oryza sativa found that there was no clear distinction between groups I and II; therefore, rice XTHs were divided into only two groups: group I/II and group III (Eklof & Brumer, 2010). The XTH members in group III could be further divided into two subgroups (IIIA and IIIB) according to their three-dimensional structures (Baumann et al., 2007; Fu, Liu & Wu, 2019). Moreover, a small outlier group was identified close to the root of the tree and was named the ancestral group. The XTHs of group I/II and group IIIB showed primarily or only XET activity, while the XTHs in group IIIA mainly displayed XEH activity (Eklof & Brumer, 2010; Nomchit et al., 2010; Opazo et al., 2017). Further studies revealed that each type of enzyme activity was determined by several structural characteristics. For example, in the protein structure of TmNXG1, loop two is the key structure affecting hydrolysis and transglycosylase activity (Mark et al., 2009). PttXET16-34 contained an important N-glycan structure, which is found in all group I/II members but absent in almost all IIIA groups (such as TmNXG1). Interestingly, the N-glycosylation site shifts from the C-terminus to the other side of the active-site cleft in group IIIB (Mark et al., 2009; Eklof & Brumer, 2010).

Increasing evidence has revealed that XTHs are instrumental for coping with abiotic stresses through cell remodeling and enhanced cell wall biogenesis in plants (Eklof & Brumer, 2010; Tiika et al., 2021). For instance, the constitutive expression of CaXTH3 has been verified to enhance resistance to salinity and drought stress in tomato plants (Choi et al., 2011). AtXTH11, AtXTH29, and AtXTH33 were observed to be upregulated through different secretory pathways in Arabidopsis seedlings treated with heat stress and drought stress (Caroli, Manno & Lenucci, 2021). A recent study revealed that the overexpression of persimmon DkXTH1 promotes tolerance to salt and drought stress by improving photosynthesis and reducing lipid peroxidation (Han et al., 2017). Additionally, transgenic tobacco with estradiol-inducible expression of SlXTH10 shows stronger growth under salinization and hypothermia conditions (Norbert et al., 2020), and GmXTH expression levels have been reported to be significantly associated with flooding stress (Li et al., 2018). Transgenic soybeans overexpressing AtXTH31 also exhibit increased tolerance to flooding stress (Li et al., 2018). Moreover, an AtXTH19 mutant was demonstrated to show lower freezing tolerance during cold and subzero acclimation than the wild type, which is likely related to differences in the cell wall composition and structure (Daisuke et al., 2020).

Taken together, the above studies highlight the essential functions of XTHs in resisting abiotic stress. In fact, the identification of novel genes involved in abiotic stress resistance and their application in genetic breeding is now considered an effective approach for the improvement of stress resistance in grapes. The existence of a high-quality de novo-assembled grape genome has made it possible to identify gene families in this species. In this study, we isolated and identified the XTH family members from grapevine and performed a complete bioinformatics analysis of the XTH family. Interestingly, we identified some putative members with potential functions under abiotic stresses, especially salt and drought stress. These findings allow in-depth research on the potential functions of the selected VvXTHs in grapevine.

Materials and Methods

Identification and biochemical analysis of XTHs in grapevine

The sequence annotations of the whole genome and the gene GFF3 file were downloaded by using CRIBI v2.1 (https://urgi.versailles.inra.fr/Species/Vitis/Annotations) (Canaguier et al., 2017). We also downloaded hidden Markov models (PF00722 and PF06958) of the XTH domain from the Pfam database (http://pfam.xfam.org) and obtained the candidate gene sequence numbers of the grapevine XTH family with HMMer software (Potter et al., 2018). To avoid duplication and the inclusion of sequences without XTH family domain characteristics, sequences without the XTH domain and sequences showing alternative splicing were removed. The EMBL-EBI online tool (http://pfam.xfam.org/search/sequence) (Gaia et al., 2021) was used to further analyze secondary structure domains, and the sequences without typical XTH domains were removed.

The relative molecular weight (MW), hydrophilicity (GRAVY), and isoelectric point (pI) of these VvXTHs were predicted and analyzed using ExPASy (https://www.expasy.org/) (Duvaud et al., 2021). Single peptide (SP) prediction was performed on the SignalP v4.1 server (https://services.healthtech.dtu.dk/service.php?SignalP-5.0).

Gene structures were analyzed with Gene Structure Display Server software (http://gsds.gao-lab.org/) (Hu et al., 2015). Conserved motifs in VvXTHs were statistically identified with the online Multiple EM for Motif Elicitation (MEME) software (https://meme-suite.org/meme/tools/meme) (Bailey et al., 2009), and TBtools was then used for the clustering and visualization of VvXTHs in grapevine. Multiple protein sequence alignments were performed with ClustalX software and the Espript 3.0 online program (https://espript.ibcp.fr/ESPript/ESPript/) (Larkin et al., 2007).

Phylogenetic analysis of VvXTHs in grapevine

To investigate the phylogenetic relationship of VvXTHs, the 34 VvXTH protein sequences from grapevine and the 33 AtXTH protein sequences from Arabidopsis were used for multiple sequence alignment by using the Clustal W program within MEGA 11.0 software (Sudhir et al., 2018). The phylogenetic tree was built using the neighbor-joining (NJ) method with 1,000 bootstrap replications and the p-distance model and was then validated by the maximum likelihood method. To better visualize the phylogenetic tree, the final tree diagram file (*.nwk) was uploaded from MEGA to Figtree and EVOLVIEW online software (http://www.evolgenius.info/evolview/) (Balakrishnan et al., 2019).

The Grape Genome Browser (12X) (http://www.genoscope.cns.fr/externe/GenomeBrowser/Vitis/) provided chromosomal location data for all VvXTHs. We used TBtools to identify and illustrate the distribution of genes on chromosomes. MCScanX with the default parameters was applied to identify gene duplication events. The CIRCOS program (https://github.com/CJ-Chen/TBtools) was used to analyze syntenic relationships among VvXTHs. VvXTHs falling within the identified collinear blocks were regarded as segmental events, and any two genes separated by a distance of less than 100 KB whose similarity exceeded 75% were considered tandem duplications. To visualize the synteny relationships of orthologous XTHs derived from grapes and Arabidopsis, Dual Synteny Plotter software (https://github.com/CJ-Chen/TBtools) was applied to construct a syntenic analysis map (Xie et al., 2018). The Arabidopsis sequences were obtained from The Arabidopsis Information Resource (TAIR) database (https://www.arabidopsis.org/) (Han et al., 2013). TBtools software was used to calculate the nonsynonymous (Ka) and synonymous (Ks) substitution rates and Ka/Ks ratio of each gene pair. Divergence times were calculated as follows: T = Ks/2 λ (λ = 6.5 × 10−9 for grapevine) (Li et al., 2019a).

Cis-element analysis of XTHs in grapevine

The sequences within 1,500 base pairs (bp) upstream of the starting codon of the VvXTHs were obtained from Ensembl Plants (http://plants.ensembl.org/index.html) as the promoter regions (Dan et al., 2017). The cis-elements were predicted with PlantCARE Web Tools (http://bioinformatics.psb.ugent.be/webtools/plantcare/html/) (Magali et al., 2002) and New PLACE Web Tools (https://www.dna.affrc.go.jp/PLACE/?action=newplace). TBtools was used to draw heatmaps and build clustering trees.

Gene expression analysis of XTHs in different grapevine organs and tissues

To understand the spatial and temporal expression patterns of VvXTHs during development, a high-throughput microarray data, from a gene expression atlas generated from different organs/tissues at different developmental stages (Marianna et al., 2012), was employed for further analysis. According to the gene ID, the expression profiles of VvXTHs was extracted from the GSE36128 data set, and we then normalized the average expression value of each gene in 54 samples (including green and woody tissues and organs at different developmental stages as well as specialized tissues such as pollen and senescent leaves). TBtools was used to draw heatmaps and build clustering trees.

To verify the reliability of the results obtained from the GSE36128 data set, the organ-specific expression patterns were examined with quantitative real-time RCR (qRT-PCR) using the five different organs (tendril, root, stem, leaf and flower) from 5-year-old trees of grapevine “Crimson” growing at the experiment station of Shandong Agricultural University (Taian, Shandong, China).

RNA extraction and expression analysis of VvXTHs

The tissue culture seedlings of Vitis vinifera cv “Crimson” seedless were grown on 1/2 Murashige and Skoog (MS) solid medium with 0.2 mM indole-3-butytric acid (IBA) under a 16-h-light/8-h dark cycle at 24 ± 1 °C for 6 weeks. Six-week-old seedlings, which transcription level changes more pronounced, were transferred to liquid medium containing 200 mM NaCl or 200 mM mannitol for salt and drought stress treatments, respectively. The treated seedlings were extracted and separated into leaves and roots for 0, 3, 6, 9, 12, and 24 h upon treatment, immediately frozen in liquid nitrogen and stored at −80 °C for RNA extraction. For each sample, three biological replicates were collected.

Total RNA was extracted from the samples treated with NaCl and mannitol using a HiPure HP Plant RNA Mini Kit (Magen, Guangzhou, China) based on the supplier’s instructions. Subsequently, first-strand cDNA was synthesized from total RNA with the PrimeScript™ RT reagent kit with gDNA Eraser (Vazyme Biotech Co., Nanjing, China). qRT-PCR was performed using a SYBR® PrimeScript™ RT-PCR Kit (TaKaRa, Dalian, China) according to the supplier’s instructions with a CFX96TM Real-Time PCR Detection System. Gene expression levels were normalized against the average expression of the internal reference gene, and the baseline and Ct (threshold cycles) value were automatically determined by the CFX Manager software program. The relative expression levels of VvXTHs were calculated using the 2−ΔΔCt comparative Ct method. The internal reference gene used in this study was Vvβ-actin7 (XM_034827164), which has been proved to be a most stable gene for normalization by comparison with other reference gene (Vvβ-actin101: XM_002265440) (Fig. S1). All experiments were performed with three biological replicates, and all the primers used in this study are listed in Table S1. To visualizing the relative difference, the expression level of 0 h treatments for salt and drought stresses and tendril for plant tissue specificity was set as 1, respectively. Then, TBtools was used to draw a heatmap for visualization.

Results

Identification and analysis of VvXTHs in grapevine

Forty-two sequences were identified by searching for two domains (Pfam: PF00722 and PF06955) with the HMMer program. We deleted six alternative splicing sequences and two sequences without typical XTH domains. As a result, we finally identified 34 VvXTHs. Described by previous studies (Cao et al., 2016; Fu, Liu & Wu, 2019; Li et al., 2018; Wan et al., 2014), we named these genes according to their chromosomal locations and named them VvXTH1-VvXTH34.

The analysis of the physical and biochemical data of the 34 VvXTHs, including their amino acids (AAs), MWs, SPs, pIs, total average hydrophilicity (GRAVY) and subcellular localization, revealed that they contained 251~369 AAs. The MW ranged from 28.5 to 41.7 kDa, while the pI ranged from 4.61 to 9.45. All XTHs exhibited hydrophilicity. Subcellular location prediction results showed that most of the genes are localized in the plasma membrane (29), while a few were localized extracellularly (5), including VvXTH10 and VvXTH12 in group IIIA and VvXTH2, VvXTH32, and VvXTH33. The majority of the proteins (80%) contained signal peptides, which were approximately 25-AA long (Table 1).

Table 1 Molecular characteristics of VvXTHs in grapevine.

Name	Gene identifier	AA	MW (Da)	SP	pI	GRAVY	Subcellular localization	
VvXTH1	VIT_201s0011g06250	279	32,099.88	–	6.60	−0.649	Plasma membrane	
VvXTH2	VIT_201s0026g00200	313	35,198.85	24	6.83	−0.296	Extracellular	
VvXTH3	VIT_201s0150g00460	307	35,270.14	35	8.65	−0.366	Plasma membrane	
VvXTH4	VIT_202s0012g02220	341	38,867.80	–	8.99	−0.374	Plasma membrane	
VvXTH5	VIT_203s0088g00650	295	34,401.83	25	7.12	−0.372	Plasma membrane	
VvXTH6	VIT_205s0062g00240	281	32,143.11	24	9.22	−0.389	Plasma membrane	
VvXTH7	VIT_205s0062g00250	279	32,239.31	24	9.07	−0.449	Plasma membrane	
VvXTH8	VIT_205s0062g00480	281	32,088.01	24	9.08	−0.406	Plasma membrane	
VvXTH9	VIT_205s0062g00610	281	32,173.18	24	9.14	−0.408	Plasma membrane	
VvXTH10	VIT_206s0061g00550	291	32,696.72	18	5.74	−0.438	Extracellular	
VvXTH11	VIT_207s0185g00050	280	32,102.92	19	7.11	−0.555	Plasma membrane	
VvXTH12	VIT_208s0007g04950	293	33,761.16	18	9.45	−0.457	Extracellular	
VvXTH13	VIT_210s0116g00520	300	34,816.90	27	4.61	−0.584	Plasma membrane	
VvXTH14	VIT_210s0003g02440	294	33,673.17	27	9.44	−0.375	Plasma membrane	
VvXTH15	VIT_210s0003g02480	290	32,860.23	27	8.18	−0.278	Plasma membrane	
VvXTH16	VIT_211s0016g03480	291	33,246.42	17	8.24	−0.338	Plasma membrane	
VvXTH17	VIT_211s0052g01180	321	36,502.43	–	4.81	−0.596	Plasma membrane	
VvXTH18	VIT_211s0052g01190	369	41,704.56	–	6.36	−0.454	Plasma membrane	
VvXTH19	VIT_211s0052g01200	297	32,951.68	29	5.22	−0.392	Plasma membrane	
VvXTH20	VIT_211s0052g01220	285	31,821.45	19	5.92	−0.404	Plasma membrane	
VvXTH21	VIT_211s0052g01230	278	31,187.75	–	5.13	−0.405	Plasma membrane	
VvXTH22	VIT_211s0052g01250	312	35,059.55	–	8.42	−0.338	Plasma membrane	
VvXTH23	VIT_211s0052g01260	296	32,919.61	26	5.07	−0.393	Plasma membrane	
VvXTH24	VIT_211s0052g01270	297	33,155.89	29	4.97	−0.361	Plasma membrane	
VvXTH25	VIT_211s0052g01280	296	32,939.66	26	5.63	−0.420	Plasma membrane	
VvXTH26	VIT_211s0052g01300	280	31,322.79	26	5.37	−0.419	Plasma membrane	
VvXTH27	VIT_211s0052g01310	269	30,176.71	29	5.69	−0.275	Plasma membrane	
VvXTH28	VIT_211s0052g01320	296	33,018.79	26	5.93	−0.425	Plasma membrane	
VvXTH29	VIT_211s0052g01330	262	29,508.94	18	5.62	−0.459	Plasma membrane	
VvXTH30	VIT_211s0052g01340	272	29,935.10	29	4.96	−0.457	Plasma membrane	
VvXTH31	VIT_212s0134g00160	296	33,657.80	27	5.60	−0.320	Plasma membrane	
VvXTH32	VIT_215s0048g02850	322	37,077.07	23	6.11	−0.346	Extracellular	
VvXTH33	VIT_216s0100g00170	251	28,460.18	–	6.65	−0.232	Extracellular	
VvXTH34	VIT_217s0053g00610	284	32,329.43	21	5.50	−0.361	Plasma membrane	
Note:

AA, amino acid; MW, molecular weight; SP, signal peptide; pI, isoelectric point; GRAVY, total average hydrophilicity.

Phylogenetic analysis and classification of VvXTHs

To investigate the evolutionary relationships and functional associations of VvXTHs with AtXTHs, we built a phylogenetic tree utilizing the protein sequences of XTHs from Vitis vinifera and Arabidopsis (Fig. 1). The VvXTHs were grouped according to the previous grouping method applied for AtXTHs and the evolutionary relationship between grapes and Arabidopsis. The results of the phylogenetic analysis indicated that the 34 VvXTHs could be divided into three groups, including 27 VvXTHs in group I/II, two in group IIIA, and five in group IIIB. In addition, one XTH protein (VvXTH11) was classified into the original ancestral group. Group I/II contained most of the members, and substantial similarity could be observed between some members of the group. The termini of the phylogenetic tree branch showed a total of 22 sister pairs, eight of which were orthologous pairs between Arabidopsis and grapevine, and six were grape homolog gene pairs. This analysis revealed that the number of VvXTHs was slightly expanded in comparison to the number of XTHs in Arabidopsis.

Figure 1 Phylogenetic analysis of XTHs of Arabidopsis and grapevine.

The amino acid-based phylogenetic tree was generated using MEGA11.0 software via the neighbor-joining method. Bootstrap test results are indicated in the tree. The different colored branches and arcs represent Group I/II, IIIA, IIIB, and the Ancestral Group, and the blue five‐pointed star represents AtXTH family members. The red triangle represents VvXTH family members.

Thirty-four VvXTHs were unevenly distributed on 13 chromosomes. In particular, Chr.11 contained the largest number of VvXTHs (15), whereas other chromosomes contained considerably fewer genes. For example, a total of 4, 3, and 2 genes were located on Chr.5, Chr.10 and Chr.1, respectively. In addition, Chr.2, Chr.3, Chr.6, Chr.7, Chr.8, Chr.12, Chr.15, Chr.16, and Chr.17 each contained only one gene (Fig. 2A). Therefore, it can be inferred that there should be no observable association between the number of XTHs and the length of chromosomes. Furthermore, the genes located on Chr.11 and Chr.5 were closely clustered together. According to the chromosome location and genome annotation information, a total of 81 tandem duplicate gene pairs were obtained (Fig. 2A). To determine the relationships among VvXTH members, we performed a collinearity analysis and found no VvXTH within the identified collinear blocks, which indicated that segmental duplication is not involved in VvXTH expansion (Fig. 2B). These results prove that the expansion of VvXTHs, especially group I/II gene members, was driven by tandem duplication. We also traced the duplication time of VvXTHs by analyzing their Ka, Ks and Ka/Ks ratio. The Ka/Ks ratios of all VvXTHs were less than 1, ranging from 0.07 to 0.28. The duplication times of all VvXTHs were also calculated. The duplication times ranged from 2.91–78.39 Mya (million years ago) (Table S2). To further evaluate the evolution and development of the VvXTH family, we constructed a comparison diagram of grapes and Arabidopsis. Eight VvXTHs were shown to be synonymous with XTHs of Arabidopsis. Among these genes, VvXTH10 is collinear with AtXTH31 and AtXTH32 in Arabidopsis, while VvXTH1 is collinear with AtXTH27 and AtXTH28 in Arabidopsis (Fig. 2C). It is speculated that there may be functional redundancy among these genes, which implies that they have important roles in evolutionary progress.

Figure 2 Systematic analysis of VvXTHs in grapevine.

(A) Thirty-four VvXTHs were mapped on grape chromosomes based on their physical positions. Eighty-one tandemly duplicated gene pairs are indicated by red lines. The scale on the left is in megabases (Mb). (B) Schematic representations of the chromosomal distribution and interchromosomal relationships of VvXTHs. Gray lines indicate all synteny blocks in the grape genome. Gene IDs on the chromosomes indicate gene physical positions. (C) Gray lines in the background indicate the collinear blocks identified in grape and Arabidopsis, while the different colored lines highlight the syntenic XTH gene pairs.

Gene structural and multiple sequence alignment analysis of VvXTHs

Gene structural analysis revealed that the closely related genes within this subfamily are characterized by a similar structure, which can be further verified by the results of phylogenetic analysis (Fig. 3A). With the exception of VvXTH32, which lacks any introns, all other VvXTH members contain 2~4 different introns. In particular, group I/II contains a large number of members, and most members have two introns. The gene sister pairs, including VvXTH23/26, VvXTH25/28, VvXTH14/15, and VvXTH8/9 at the terminal branch of the evolutionary tree, have highly similar exon/intron structures. In addition, compared with the adjacent gene VvXTH27, VvXTH24 has lost an exon and exhibits different intron and exon lengths. The members of group IIIA all have three introns and four exons and present high structural similarity. The members of group IIIB have developed different numbers and lengths of intron/exon structures during the long evolutionary process (Fig. 3C). In general, most VvXTHs present the same intron/exon structure pattern and remain conserved during evolution, which is consistent with the results obtained in other plants.

Figure 3 Phylogenetic relationships, structures and conserved motifs of VvXTHs.

(A) Phylogenetic tree inferred from the protein sequences of VvXTHs. Branch colors represent different groups. (B) The motif composition of the VvXTHs identified using MEME. The different colored boxes represent different motifs and their positions in each VvXTH sequence. Each motif is indicated by a colored box in the legend at the bottom. (C) Gene structure of VvXTHs. The boxes represent exons or untranslated regions (UTRs), and lines represent introns. (D) Schematic representation of the conserved domains found in grape. (E) Multiple sequence alignments of the conserved domains of the VvXTHs. The black lines indicate the conserved domains. N-glycosylation residues are indicated with asterisks.

Based on the results of MEME motif analysis, motif three and motif four are highly conserved in all VvXTHs. Motif three is a characteristic domain that catalyzes enzymatic contact reactions, which denoted as (D/N)E(I/L/A/V/F)(D/T)(F/I)E(F/I/L)LG (Figs. 3B and 3D). Among these, the first glutamate residue (E) indicates the catalytic nucleophile that initiates the enzymatic reaction, and the second E residue functions represents a base to activate the entrant substrate. In addition, all members except for VvXTH30 contain motif one; moreover, members of the same group share a similar motif composition (Fig. 3B). For instance, motif two only exists in group I/II, and motif eight only exists in group IIIA and IIIB. Genes in the same clade, especially those that are closely related, such as (1) VvXTH23, VvXTH26, VvXTH25, and VvXTH28; (2) VvXTH6, VvXTH7, VvXTH8, and VvXTH9; and (3) VvXTH10 and VvXTH12, can share much more similar motif structures (Fig. 3B). In addition, motif seven, motif nine, and motif 10 only exist in group I/II, and most of the group members contain the motifs mentioned above (Fig. 3B). The members of group IIIA contain five motifs with the same distribution. Group IIIB members contain 4–6 motifs; while they all share four identical motifs, only VvXTH4 and VvXTH32 have motif six, and only VvXTH33 does not have motif five. The results of multiple sequence alignment also confirmed that the conserved domain active site is present in all VvXTHs. Moreover, with the exception of VvXTH2 (IIIB), VvXTH10 (IIIA), and VvXTH12 (IIIA), potential N-glycosylation residues are located near the active site in the 31 other VvXTHs (Fig. 3E). Conserved domain predictions suggested that members of the same subfamily may have similar structures and may be involved in similar functions. Thus, we need to focus on distinctive members that may present surprising functions that remain to be discovered.

Organ-specific expression pattern analysis of VvXTHs

Through the expression profile (GSE36128) analysis of the GEO data set, we obtained the specific expression patterns of VvXTH members in different organs and developmental periods of grapevine to predict the functions of VvXTHs in growth and development (Fig. 4). According to the results of cluster analysis, the VvXTH families were classified into four groups (A–D): group A contains seven genes with high expression levels in berry peels, skins, shafts, and tendrils; group B includes four members with high expression levels only in stems and tendrils but low expression in other organs; group C includes eight genes with very low expression levels in all organs; and group D is the largest (15 members) subfamily and shows the highest expression in berries, shafts, and tendrils. In addition, the VvXTHs had higher expression levels in the pulp, peel, and stem during the V (veraison), MR (mid-ripening), and R (ripening) periods, indicating that VvXTHs may be related to fruit ripening. In short, the four groups of VvXTHs present specific expression profiles depending on the organ and developmental stage. This interesting phenomenon may be due to the specific functions of these specific genes in related tissues.

Figure 4 Expression patterns of VvXTHs in different organs and developmental stages.

Rows represent VvXTH members, while columns represent different developmental stages and organs. The expression levels of VvXTHs are indicated by the intensity of color. The phylogenetic tree on the left side of the heatmap is based on the hierarchical clustering of the expression profiles of VvXTHs in 54 samples.

To verify the reliability of the organ-specific expression profiles, qRT-PCR analysis was conducted on five different tissues (tendril, root, stem, leaf and flower) of grapevine “Crimson” for VvXTHs, then the qRT-PCR results were compared with the data obtained from the GSE36128 data set of the cultivar “Corvina” (Marianna et al., 2012) with the same tissues at the corresponding developmental stages. It was found that the expression patterns of VvXTHs were generally consistent with the data obtained from the GSE36128 data set (Fig. S2), which suggests that temporal and spatial expression of VvXTHs is generally similar in different cultivars, even grown in different conditions.

Transcriptional profiles of VvXTHs under abiotic stress

The PlantCARE database and New PLACE database were utilized to identify cis-elements in the DNA sequences 1.5 KB upstream of the VvXTH start codons. The results showed that all 34 VvXTHs contained a variety of abiotic and biotic stress response elements, phytohormone response elements, and growth and development-related response elements (Fig. 5A). Similarly, in the New PLACE database, all 34 VvXTHs were predicted to contain phytohormone response elements and elements involved in the responses to a variety of abiotic stresses, including cold and heat, ABA, dehydration and salinity (osmotic) stress (Fig. 5B). As shown in Fig. S3, two drought stress response elements (MYB and MYC) exist in almost all members, and 80% of the gene members contain defense and stress response elements (STREs), which indicates that the VvXTH family probably shows important functions when plants are subjected to abiotic or biotic stress. Abscisic acid response elements (ABREs) and salicylic acid response elements (TCA elements) are abundantly present in VvXTH family members, which indicates that the VvXTH family may also be involved in hormone regulation during plant growth and the response to stress. The presence of a meristem development control element (CAT-box) in most VvXTH members suggests that the VvXTH family may have significant effects on the regulation of plant growth and development. In particular, the 34 VvXTHs were predicted to contain a large number of light response elements in the New PLACE database, and light-responsive elements (GATABOX and Box-4) were present in most VvXTH members, suggesting that the VvXTH family plays important roles in photosynthesis and photomorphogenesis.

Figure 5 Cis-element analysis in the promoter regions 1,500 bp upstream of the start codons of VvXTHs.

The prediction analysis was performed by using plantCARE (A) and New PLACE (B). The bar graphs represent the total number of cis-elements in each gene promoter region. Different colors represent different types of cis-elements. Three types of cis-elements were predicted in plantCARE. Five types of cis-elements were predicted in New PLACE.

Promoter analysis demonstrated the widespread presence of cis-elements associated with abiotic stress in the promoter regions of VvXTHs, revealing the possible induction of VvXTH expression by abiotic stress. To further investigate the potential roles of VvXTHs in response to abiotic stress, especially drought stress and salt stress, we selected 14 VvXTH members harboring abiotic response cis-elements for further study. Six-week-old grape seedlings were exposed to 200 mM NaCl or 200 mM mannitol, and the expression of 14 VvXTHs was examined in separated leaves and roots. In roots, the expression levels of 11 genes were upregulated under salt stress, among which four members were significantly upregulated. The expression of VvXTH5, VvXTH20, and VvXTH34 was increased by more than two-fold, and that of VvXTH4 was increased by more than four-fold. Interestingly, the expression of VvXTH4 peaked after 9 h of treatment, presenting an obviously different pattern from the other genes. This shows that these genes may respond to salt stress in different ways. Under drought stress, most of the genes whose expression was upregulated reached a peak after 3 h of treatment, and some genes were upregulated by more than four-fold (VvXTH3 and VvXTH20). VvXTH10 expression reached a peak after 12 h of treatment, indicating that this gene may be expressed at a later time. The number of upregulated VvXTHs in leaves relative to roots decreased after stress treatment, but these genes were more highly upregulated. Among these genes, VvXTH3, VvXTH10, and VvXTH31 were upregulated by approximately ten-fold. The above genes might play particularly crucial roles in the leaf response to salinity stress. Taken together, our findings indicate that the expression of VvXTHs could be altered by salt and drought stress, suggesting that VvXTHs may participate in reactions to abiotic damage, especially under salt and drought stress.

Discussion

The XTH family consists of modification enzymes that can rebuild cell walls by modulating the construction and composition of xyloglucan cross-links (Campbell & Braam, 2010). According to previous studies, various members of this family have been identified in Arabidopsis thaliana, Oryza sativa, Medicago truncatula, Nicotiana tabacum, Solanum lycopersicum, and Ananas comosus, and these proteins have been verified to play critical roles in development, biotic stress, and abiotic stress (Meng et al., 2018; Li et al., 2019b; Yokoyama, 2004; Kurasawa et al., 2009; Xuan et al., 2016). The release of the most recent grape genome database made it possible to identify the grape XTH family (Ariga, Muneyuki & Yoshida, 2007). In this study, thirty-four VvXTHs were systematically identified and characterized using bioinformatics approaches. The results showed that the number of identified VvXTHs (34) (Fig. 1) was slightly greater than the numbers found in Arabidopsis thaliana (33) and Oryza sativa (29) (Yokoyama, 2004; Kurasawa et al., 2009), which may be related to pedigree-specific gains and losses as well as gene duplication events. Gene duplication is a primary driver of the expansion of gene families, and tandem duplications and segmental duplications are considered the primary duplication modes (Zhu et al., 2014). Previous studies of the XTH family have also reported gene tandem duplications or segmental duplications in barley, soybean, and tobacco (Fu, Liu & Wu, 2019; Li et al., 2018; Meng et al., 2018).

Interestingly, we observed that the thirty-four identified VvXTHs were located on 13 chromosomes, and Chr.11 and Chr.5 contained gene clusters (Fig. 2). Based on the definition of gene tandem duplication, VvXTH17-VvXTH30 and VvXTH6-VvXTH9 represent gene tandem duplication events. According to the analysis of the Ka/Ks ratio (Table S2), all genes showed ratios of less than 1, which indicates that they are under intense purifying selection (Hurst, 2002). Hence, the role of gene tandem duplication in VvXTH family expansion, particularly in increasing the number of VvXTH members and their functional diversification, is irreplaceable.

According to gene function and Clustal analyses, similar to other plants, the 34 VvXTHs are divided into groups I/II, IIIA, and IIIB and an ancestral group (Fig. 1). According to previous studies, due to the unclear distinction between groups I and II, these subgroups were combined into one group (group I/II) (Campbell & Braam, 1999). The XTHs in group IIIA mainly display XEH activity, while those of group IIIB showed obvious XET activity, suggesting a functional distinction between groups IIIA and IIIB (Eklof & Brumer, 2010; Nomchit et al., 2010; Opazo et al., 2017). Serines or threonines located near the catalytic center of XET are typical residues for N-glycosylation, and the results of multiple sequence alignment showed that the members of groups I/II and IIIB (except for VvXTH2) contain N-glycosylated residues, while those of group IIIA do not (Mark et al., 2009). Therefore, we speculated that VvXTH10 and VvXTH12 proteins in group IIIA might possess XEH activity and that VvXTH4, VvXTH32 and VvXTH33 proteins in group IIIB might show XET activity in grape, which is in agreement with previous research findings (Fig. 1) (Mark et al., 2009; Miedes & Lorences, 2009).

The analysis of gene structure is of great significance to further clarify the origins, evolution, and genetic relationships of species. XTH family members show a relatively wide variety of structures. Specifically, most members of the grape XTH family contain three or four introns (Fig. 2C), while others have fewer intron, which may be related to gene splicing (Mount et al., 2012). It is precisely because of the existence of multiple introns that gene splicing becomes more complicated, and the number of different XTH expression products increases. According to a comparison of the AA sequences of Arabidopsis thaliana, Populus tomentosa, Hordeum vulgare, Brassica rapa, and Brassica oleracea, even when the difference in protein size is obvious, the active-site domain is still conserved in the reported XTH proteins (An et al., 2017). In this study, all 34 VvXTHs were found to contain motif three (Opazo et al., 2017), which suggests that XTH proteins may play similar roles in the plant kingdom. It has been reported that the active site mediates catalytic activity, which can catalyze hydrolase activity and carry out cell wall structural modification and rearrangement by cutting and repolymerizing cellulose single chains (Li et al., 2018; Behar, Graham & Brumer, 2018). The cross-linked xyloglucan structure has critical functions in maturation and resistance to abiotic stress (Bulone, Schwerdt & Fincher, 2019).

Previous studies have shown that XTHs are of vital importance in plant resistance to abiotic stress (Chen et al., 2019; Dong et al., 2019; Li et al., 2019b). The expression of CaXTH3 is induced by a variety of abiotic stresses, such as drought, high salt, and low temperature, and the tolerance of CaXTH3 transgenic tomato plants to salt and drought stress is thereby significantly improved (Choi et al., 2011). Additionally, the heterologous expression of PeXTH in tobacco improves plant osmotic tolerance by reducing water loss and reducing the speed of stomatal opening (Han et al., 2014). To study the potential function of VvXTHs against abiotic stress, we carried out promoter analysis and tissue expression analysis (Figs. 3 and 4). The results indicated that the upstream promoter regions of almost all members of the grape XTH family contain MBS, MYB, MYC, and ARE cis-elements for responding to drought stress. Furthermore, 47% of these sequences contain ABREs, to respond to ABA (Fig. 4). Under drought conditions, ABA inhibits root growth and development, represses seed germination, and promotes the shedding of senescent tissues and organs (Hirayama & Shinozaki, 2007). Some VvXTHs exist in the mature stage, and ABA may promote XTH expression and affect organ abscission (Figs. 3 and 6). In addition, a few members of the VvXTH family also contain DRE action elements, which means that XTHs can potentially respond to salt stress, in addition to drought stress (Fig. 5). It is also interesting that the expression of VvXTHs varies in different organs. Genes from the same gene cluster of gene tandem repeat events, such as VvXTH6 and VvXTH7 or VvXTH26, and VvXTH30, show differences in expression in organs at different stages. These results implied that during evolution, closely related genes have undergone subfunctional evolution, functionalization or nonfunctionalization, helping grapes adapt to a variety of growth environments. The expression profiling of VvXTHs under different stresses revealed the induction of VvXTH3, VvXTH31, and VvXTH10 in roots.

Figure 6 Expression profiles of VvXTHs under abiotic stress.

Heatmap showing the relative expression of 14 VvXTHs, detected by qRT-PCR, in roots and leaves of 6-week-old “Crimson” grape seedlings after treatment with 200 mM NaCl and 200 mM mannitol for 0, 3, 6, 9, 12, and 24 h (0 h treatment as the control). Experiments were performed in biological triplicates.

Taken together, these findings provide novel information about VvXTHs under abiotic stress, especially drought and salt stresses. It can be speculated that the above genes may show increased cell wall-related functions under stress by combining with xyloglucan. Nevertheless, further molecular and genetic identification efforts are needed to verify their functions.

Conclusions

In this study, 34 XTHs, which could be further divided into group I/II, group IIIA, and group IIIB, were successfully isolated and identified in grapevine. It was shown that the VvXTHs are unevenly distributed on 13 chromosomes. According to collinearity analysis, tandem duplication of genes may have occurred on Chr.5 and Chr.11. Furthermore, all VvXTHs contain conserved XTH domains and active sites. Expression analysis showed that some VvXTHs can effectively respond to salt and drought stress at the transcriptional level. In this context, the results of the present investigation will lay a foundation for future investigations of the function of VvXTHs.

Supplemental Information

Supplemental Information 1 Raw data.

Click here for additional data file.

Supplemental Information 2 Supplementary Figures and Tables.

Click here for additional data file.

Additional Information and Declarations

Competing Interests

Author Contributions

Data Availability

The authors declare that they have no competing interests.

Tian Qiao conceived and designed the experiments, performed the experiments, analyzed the data, prepared figures and/or tables, authored or reviewed drafts of the article, and approved the final draft.

Lei Zhang conceived and designed the experiments, performed the experiments, analyzed the data, prepared figures and/or tables, authored or reviewed drafts of the article, and approved the final draft.

Yanyan Yu performed the experiments, prepared figures and/or tables, and approved the final draft.

Yunning Pang performed the experiments, prepared figures and/or tables, and approved the final draft.

Xinjie Tang analyzed the data, prepared figures and/or tables, and approved the final draft.

Xiao Wang analyzed the data, prepared figures and/or tables, and approved the final draft.

Lijian Li analyzed the data, prepared figures and/or tables, and approved the final draft.

Bo Li conceived and designed the experiments, authored or reviewed drafts of the article, and approved the final draft.

Qinghua Sun conceived and designed the experiments, authored or reviewed drafts of the article, and approved the final draft.

The following information was supplied regarding data availability:

The raw data are available in the Supplemental File.

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
