# Peer review of "Identification and expression analysis of xyloglucan endotransglucosylase/hydrolase (XTH) family in grapevine (Vitis vinifera L.)"

_PeerJ, doi:10.7717/peerj.13546_

## Round 0.1 · original submission · Major Revisions

The authors are advised to do the corrections as suggested by reviewers.

Reviewer 1 ·

Basic reporting

The manuscript "Identification and expression analysis of xyloglucan endotransglucosylase/hydrolase (XTH) family genes in grapevine (Vitis vinifera L.)" reports the genome-wide identification of XTH genes and expression profiling in grapes.
XTH genes are known to play important role in plant development as well as response to abiotic and biotic stresses. Therefore, this MS will be of interest to grape researchers.
The manuscript is well written. Sufficient background information is given.

Experimental design

The experimental design is appropriate. The methodology is given in sufficient detail.
The results have been given in detail covering all the aspects. All the relevant figures and tables have been included, the figure quality is good. The relevant raw data has been shared.

Validity of the findings

The conclusions are well stated and linked to the original research question.

Reviewer 2 ·

Basic reporting

1.) The manuscript needs language improvisation to address issues like the following:
Line 46-47 “Grapevine (Vitis vinifera L.) as one of the most economically fruit crops is globally widespread cultivation (Zhu et al., 2019).”
Line 65-66 "The former hydrolyses β-1, 4 glycosidic bonds of xyloglucan to form the cleavage and connection of xyloglucan chains."
2.) The article provides sufficient background information on the existing problem as to why VvXTH family genes family identification and characterization can be important in stresses ad developmental stages.
3.) Article is properly supplemented with figures and tables.

Here are few additional comments for the abstract and introduction sections, that need to be taken care of:
Abstract:
1. In order to avoid redundancy, eliminate "gene/genes" after XTH in abstract and text throughout.
Introduction:
1. Line 46-47 “Grapevine (Vitis vinifera L.) as one of the most economically fruit crops is globally widespread cultivation (Zhu et al., 2019).” This is an awkward line and needs rephrasing.

2. Line 48-51 “The growth of grapes in natural environment inevitably suffers from a series of abiotic pressures from salinity, drought and extreme temperatures, which changes the morphology of the plant, disrupts the biofilm system and causes oxidative damage, and ultimately affects the quality and yield of the fruit”.

3. Please include citations for the information presented in the following lines:
Line 48-51“The growth of grapes in natural environment inevitably suffers from a series of abiotic pressures from salinity, drought and extreme temperatures, which changes the morphology of the plant, disrupts the biofilm system and causes oxidative damage, and ultimately affects the quality and yield of the fruit.”

4. Line 77-81 “Further studies found that each enzyme activity was determined by several structural characteristics. For example, in the protein structure of TmNXG1, loop2 is the key structure affecting hydrolysis and transglycosylase activity. PttXET16-34 has an important N-glycan structure, which is present in all group I/II members and absent in almost all III-A groups (such as TmNXG1).”

5. Line 55 “Xyloglucan Endotransglucosylase/Hydrolase (XTH), a cell wall that modifies enzyme” is not clear. Did you mean XTH, a cell wall modifying enzyme?

6. A statement clearly indicating the novelty of the study should be included in the Introduction section.

Experimental design

In the present manuscript, the authors have explored the genome of Vitis vinifera to identify XTH genes. Detailed bioinformatics analysis was conducted to study the characteristics like their chromosomal location, gene and protein structure, the subcellular location of proteins, phylogenetic classification. The role of various cis-regulatory elements in the promoters well supports the regulatory roles in abiotic stress and development. Further, the authors have also conducted expression studies of the VvXTH family in drought and salt stresses and developmental stages.
The work has laid down the basic groundwork for future validations of these genes and their utilization in developing resistant/tolerant grapes.
Comments in the experimental design that need to be addressed:
1. Line 122 “The single peptide (SP) was predicted by SignalP v4.1 server”. Single peptide/signal peptide?

2. Proper references/citations for software/tools used to carry out various bioinformatics analyses of the XTH gene family are missing from the manuscript.

3. Define an abbreviation at its first instance; thereafter, use the abbreviated form consistently throughout the text. For instance: in Line 68, At (Arabidopsis thaliana) should be defined.

4. Line 149: “Gene structures were performed with the Gene Structure Display Server (GSDS:http://gsds.cbi.pku.edu.cn/) software”. The use of correct terminology is important in creating a good influence of the manuscript on the readers. “Performed” should be replaced with either analyzed or a suitable word.

5. Computational prediction of Cis-regulatory element has very less reliability since most of the motifs are too short (4-6 bases) therefore random occurrence will be very high. I suggest confirmation with at least 2-3 prediction tools to reduce the errors/random occurrences. Or if the authors can prove the regulatory roles of a few selected Cis-elements through wet-lab experiments.

6. I suggest rewriting the Sub-heading "Organ-specific expression analysis of VvXTH gene family". As to how the gene expression was normalized is not clear.

6. Line 276: “In addition, during the V, MR, and R periods, the VvXTH genes”. What do V, MR, and R periods stand for? Clearly explain?

7. Why was qRT-PCR verification for VvXTH not carried out in different developmental stages?

Validity of the findings

1. The authors in the manuscript have mentioned the amino-acid composition of motif 3 i.e “DEIDFEFLG”, which is a conserved motif. But there are certain substitutions across VvXTH protein sequences as shown in Fig: 4e. How can the authors justify this statement?

The authors have discussed most of their findings with relevant references supporting the results.

·

Basic reporting

The idea of the article is understandable, but unfortunately not in an easy way. Indee, the article is not simple to read in part because of the English language used in part for the grammatical errors.

The introduction section doesn't inclue the most recent works on the topic (see my comments in the text).

Some ideas of the introduction section seem to lack relevance (like the wide range of applications of the grapevine) or could be further described due to the relevance for the study (the assembly or assemblies used for this study for example) considering that is a genome-wide identification of a not yet characterized gene family in grapevine.

Figures are clear and the importance or use of each one could be boosted.

Experimental design

The workflow for the identification of the genes seems to be appropriate and is similar to others in the literature, but could be further explained without increasing too much the text extension.

The stress induction experiment seems to be properly designed, and covers the expression of the selected genes in a wide range of hours which gives information of a relatively short to mid-term response. Why were these hours selected and not others?

The connection of the two parts of the study (in silico and stress analysis) is quite subtle to me, therefore I suggest to improve this part mainly in the Results section.

Validity of the findings

Although the genome wide identification of XTH has been performed already in other species, is new for grapevine and therefore could be relevant considering the reasons stated in the same article to study the family (the response and effects in drought and salt stress.)

Additional comments

1. In general the article is “readable” but in general requires a profound revision at the English language level (Starting from the very first sentence in the Introduction section). I don't go into detail, but the article to me has to be completely revised by a fluent English Speaker.

2. Citations must be consistent with the structure in all the text (lines 100, 101 and 102).

3. References must be checked (how is the order of the Journal? In order of appearance with APA format?)

4. Further important points to have in mind (which I inserted also in the pdf file as comment)

- Link to CRIBI website is out of use (and is not correctly cited)
- Indicate version of the genome and annotation used (it's possible to understand that it’s the CRIBI v2 from table 1, but I think has to be mentioned)
- As a general comment: the procedure adopted to identiy and validate the gene/protein models coding for XTH proteins in the garpevien genome has to be better explain and supported. To me teh analysis done is not clear and validated.
- Some XTH sequences were discarded but the reasons are inconsistent and not clear and need to be better explained and supported
- The adopted method for qPCR data analysis is quite obsolete. Indeed the method proposed in the last 10 years of research in based on the method described by Hellemans et al., (Genome Biology 2007, 8:R19) which essentially takes in consideration a selection of the most stable genes for normalization (which has to be identified within the experiment) and the fact that each primer pair has its own efficiency which in the most of the cases doesn't correspond to 2.
- Names of molecules and bound (b-1,3-glucans, b-1,3(4)-glucans, β-1,4 glycosidic bond) must be consistent between them.
- Is there a correspondence among phylogenetic ,co-expression in different organs /tissues and cis-elements analysis which could support the idea of a specific co-regulation of specific members having evolved together? if it is the case, add a paragraph about it and discuss this important point.
- Figures 5 and 7 could have been compared or tried to integrate (in a graphical way) as they refer (both) to gene expression.
- The selection of genes could have been performed not in a random way, for example using gene co-expression network or similar (AggGCNs, VESPUCCI (Moretto et al., 2016, Frontiers in Plant Science 7: 633 and VTCdb databases (D. C. Wong et al. 2013)), etc), or comparing the behaviour of the orthologs in other species (the ones reported, i.e. tobacco, thale cress, rice, etc) to have a reference for the selection
-The discussion in general should be based on the revised results obtained after the suggestions given, focusing on the evidences base don which some selected members have been selected as putatively involved in abiotic stresses and therefore candidate to further functional characterization.

---

## Round 0.2 · Minor Revisions

Dear Authors, Kindly go through the comments for minor revision by one of the reviewers and respond.

Reviewer 2 ·

Basic reporting

The authors have revised and incorporated suggested changes to the manuscript.

I recommend the authors, to make corrections in the following places:
1.) Italicize the gene name.
2.) The word "gene" is to not be added when writing the gene name, for example, XTH and not XTH gene.

Experimental design

1.) In the experimental section, the authors have not yet properly described as to how was gene normalization carried out.

2.) In the previous review, the authors were questioned, as to why the qPCR validation was not performed for developmental stages. The reason is still not clear.

Validity of the findings

Previously, authors were asked to review the amino-acid composition of a conserved motif, for which they have removed that line from the results section, and added a single line in the introduction supported with previous works.
I recommend that the authors should add a separate line in the manuscript about their own findings regarding the amino acid substitutions in their XTH proteins.

·

Basic reporting

No comments

Experimental design

No comments

Validity of the findings

No comments

Additional comments

No comments

---

## Round 0.3 · accepted · Accept

The manuscript has been revised extensively and is ready for acceptance.